# Association of the Combination of Moderate-to-Vigorous Physical Activity and Sleep Quality with Physical Frailty

**DOI:** 10.3390/geriatrics9020031

**Published:** 2024-03-04

**Authors:** Tsubasa Yokote, Harukaze Yatsugi, Tianshu Chu, Xin Liu, Lefei Wang, Hiro Kishimoto

**Affiliations:** 1Department of Behavior and Health Sciences, Graduate School of Human-Environment Studies, Kyushu University, Fukuoka 819-0395, Japan; tsubasayokote@icloud.com (T.Y.); chutianshu_japan@yahoo.co.jp (T.C.); rakuhi0111@gmail.com (L.W.); 2Faculty of Arts and Science, Kyushu University, Fukuoka 819-0395, Japan; haru19920424@gmail.com; 3Epidemiological Study Group, Medical Evidence Division, Intage Healthcare Inc., Tokyo 101-0062, Japan; liuxinjp1992@gmail.com; 4Center for Health Science and Counseling, Kyushu University, Fukuoka 819-0395, Japan

**Keywords:** physical activity, sleep quality, physical frailty, cross-sectional study, older adults

## Abstract

Background: The association of the individual and combined effects of moderate-to-vigorous physical activity (MVPA) and sleep quality with physical frailty in community-dwelling older adults is still unknown. Subjects and Methods: A cross-sectional study was conducted with a sample of older adults who had not required nursing care or support services. Physical frailty was assessed using Liu’s definition based on Fried’s concept. MVPA was measured by a triaxial accelerometer, and individuals who met either moderate physical activity (MPA) for ≥300 min/week, vigorous physical activity (VPA) for ≥150 min/week, or both were defined as “MVA^+^”. “SLP^+^” was defined as a Pittsburgh Sleep Quality Index score of <5.5 points. Results: A total of 811 participants were included in the final analysis. After adjusting for the multivariable confounding factors, the odds ratios (ORs) and 95% confidence intervals (CIs) for physical pre-frailty and frailty in the MVA^−^SLP^+^ (OR, 2.56; 95%CI, 1.80–3.62) and the MVA^−^SLP^−^ group (OR, 3.97; 95%CI, 2.33–6.74) were significantly higher compared with the MVA^+^SLP^+^ group. Conclusion: Community-dwelling older adults who did not meet the MVPA criteria, regardless of sleep quality, had a higher prevalence of physical frailty.

## 1. Introduction

Japan is facing a super-aged society, and the proportion of frail older adults is also on the rise. Frailty refers to a vulnerable state in which one’s physiological reserve declines, and individuals with frailty become more susceptible to stress in later life [1]. Physical frailty in particular is closely linked to the need for caregiving, and its presence increases the risk of various diseases [2,3,4]. It is thus essential to identify factors and approaches that can reduce the risk of the development of physical frailty in community-dwelling older adults.

Physical activity is a factor that has been reported to contribute most significantly to reducing the risk of physical frailty [5]. Numerous studies have explored the negative relationship between moderate-to-vigorous physical activity (MVPA) and physical frailty [6,7]. Our research has demonstrated that compared to community-dwelling older adults’ exercise habits or social participation, there is a stronger association between their MVPA and their risk of physical frailty [8]. Conversely, sleep is a critical component of daily life, and age-related declines in sleep quality have been reported. For example, Ohayon et al. observed a series of healthy adults and documented increased nocturnal awakenings and decreased deep sleep from the age of 55, indicating a relatively early decline in sleep quality [9]. Although an association between poor sleep quality and an increased risk of physical frailty has been reported, the physical activity levels of the participants were not examined [10,11,12].

Other investigations have suggested an association between poor sleep quality and slower maximal gait speed, even after adjusting for average physical activity (counts/min/day) evaluated by a triaxial accelerometer [13]. Reduced sleep quality could thus be an independent factor that increases the risk of physical frailty. A positive association between the level of physical activity and the quality of sleep has been proposed [14], but even among individuals who engage in sufficient physical activity, it is possible that some may experience poor sleep quality, and these individuals may potentially be at a higher risk of physical frailty. The combination of insufficient MVPA and poor sleep quality may also synergistically increase that risk.

The objective of this study was to determine the association of individual and combined effects of MVPA and sleep quality with physical frailty in community-dwelling older adults.

## 2. Materials and Methods

### 2.1. Study Design

The study’s design was cross-sectional. We used the baseline data obtained in the Itoshima Felix Study conducted in 2017 [15].

### 2.2. Participants

Our initial study population comprised 10,000 citizens aged 65–75 years who lived in Itoshima City, Fukuoka, Japan and responded to the Community Needs Survey in 2016. None of the citizens were certified as requiring support or nursing care under Japan’s Long-Term Care Insurance System [16]. Taking into account the size of the district, we randomly selected 5000 of the citizens and sent them invitations to participate in the Itoshima Felix study and complete questionnaires. Among these individuals, 930 completed physical function tests and additional questionnaires at local community centers. Ultimately, in the final analysis, we analyzed the data of the 811 participants who met the criteria for valid data on physical activity measured by a triaxial accelerometer and had no missing data on sleep quality and physical frailty (Figure 1).

This study was approved by the Institutional Review Board of Kyushu University (approval no. 201708), and informed consent was obtained from all of the participants.

### 2.3. Measurements

#### 2.3.1. Physical Frailty

The criteria that were used for assessing the participants’ physical frailty were determined based on the frailty phenotype, specifically focusing on values in the bottom 20% of the population [1,17]. These criteria encompass five components: weight loss, low grip strength, psychological stress, slow gait speed, and low physical activity. In the present study, we defined ‘frailty’ as meeting three or more of these five criteria, ‘pre-frailty’ as meeting one or two of the criteria, and ‘robust’ as meeting none of the criteria.

Weight loss was defined as responding “yes” to the question, “Have you experienced unintentional weight loss exceeding 2–3 kg in the previous six months?”

Psychological stress was assessed using two items from the six-item method developed by Kessler et al. [18]. Participants were asked about their feelings in the past month, specifically whether they felt that everything they did was an effort and whether they felt exhausted without any reason. Response options ranged from ‘not at all’ to ‘always’ If any of the responses fell into the categories of ‘sometimes’, ‘mostly’, or ‘always’, the participant was considered exhausted.

Low grip strength was defined based on grip strength, with individuals scoring in the lowest 20% of maximum grip strength stratified by gender and body mass index (BMI). Grip strength was measured using a Smedley grip strength meter (GRIP-D, T.K.K. 5401; Takei Scientific Instruments, Niigata, Japan) with a posture in a natural standing position where there is no bending of the shoulders or elbows while standing. Measurements were taken for both hands, alternating between hands, and repeated. The greater values from both hands were averaged [19].

Slow gait speed was identified based on gait speed, with individuals scoring in the slowest 20% of maximum gait speed determined by a 5 m walking test. In the test, the participant walked an 11 m distance from a stationary standing position, and the time taken to walk was measured at the 5 m mark between the 3 m and 8 m points. The participants were allowed to use walking aids during the measurements. Gait speed was assessed twice, and the faster speed was recorded. The results were stratified by gender and height [20].

Low physical activity was evaluated by measuring the participant’s physical activity energy expenditure (PAEE, kcal) using the aforementioned triaxial accelerometer. Low physical activity was defined as scoring in the lowest 20% of energy expenditure from physical activity per day, stratified by sex. The quantification is expressed as kilocalories per kilogram of body weight expended per day (kcal/kg/day). We introduced the accelerometer (Active Style Pro HJA-350IT, Omron Healthcare, Kyoto, Japan) to the participants. The accelerometer is a device that continuously measures exercise intensity using an algorithm derived from the relationship between data obtained by an acceleration sensor and energy consumption determined by expired gas analysis. Post-processing of the data followed established procedures from previous studies [21]. Anteroposterior (*x*-axis), mediolateral (*y*-axis), and vertical (*z*-axis) acceleration measurements were collected during each activity at a rate of 32 Hz with 12-bit accuracy. The acceleration data range for each axis is ±6 G, resulting in a resolution of 3 mG. Each signal from the triaxial accelerometer underwent high-pass filtering with a cutoff frequency of 0.7 Hz to eliminate the gravitational acceleration component. The integral of the absolute value of the acceleration signals from all three axes was calculated over 10 s intervals. The researchers explained the method of wearing the activity monitor to each participant individually, both in written instructions and verbally, and assisted the participants in putting on the device. The participants wore this accelerometer continuously for 7 days. We defined ‘valid data’ as the accelerometer data recorded consistently for ≥10 h/day on ≥4 days [22]. Accelerometer data are known to be more accurate than self-reported questionnaire estimates, and the usage of accelerometers in general populations is increasing [23]. The accuracy of the intensity estimation by the Active Style Pro accelerometer has been validated with the Douglas bag method [24]. The use of a triaxial accelerometer to assess physical activity also allows for a more precise estimation of activity intensities compared to conventional uniaxial accelerometers [24,25].

#### 2.3.2. Physical Activity

According to the Physical Activity Guidelines proposed by the World Health Organization (WHO) in 2020, it is recommended that older adults engage in moderate-intensity physical activity (MPA) for 150–300 min/week or vigorous-intensity physical activity (VPA) for 75–150 min/week [25]. To obtain even greater benefits, it is recommended that older adults engage in MPA for ≥300 min/week or VPA for ≥150 min/week [26]. Our study participants were older adults who were living independently, and we thus defined ‘meeting the MVPA criterion’ as either engaging in MPA for ≥300 min/week, VPA for ≥150 min/week, or both.

#### 2.3.3. Sleep Quality

The Pittsburgh Sleep Quality Index (PSQI) was used to assess the study participants’ sleep quality. The PSQI is a seven-item questionnaire, with each item being rated on a scale of 0 to 3, resulting in a total score ranging from 0 to 21. Higher scores indicate poorer sleep quality. Based on previous research, we used a score of ≥5.5 as the threshold to define decreased sleep quality and a score of <5.5 to define good sleep quality [27].

### 2.4. Statistical Analyses

We classified the participants into four groups: those who met the criteria for both MVPA and good sleep quality (MVA^+^SLP^+^), those who met the criteria for MVPA but had poor sleep quality (MVA^+^SLP^−^), those who did not meet the criteria for MVPA but had good sleep quality (MVA^−^SLP^+^), and those who did not meet the criteria for both MVPA and sleep quality (MVA^−^SLP^−^). The proportions of physical pre-frailty and frailty in these four groups combining MVPA status and sleep quality deterioration were compared by a one-way analysis of variance (ANOVA) or the Kruskal–Wallis test. Subsequently, multiple comparisons between groups were conducted using the Bonferroni method to investigate the differences in each item among the groups, specifically comparing the group of MVA^+^SLP^+^ with the other three groups.

We also performed an ordinal logistic regression analysis using the MVPA status and poor sleep quality as exposure factors and three categories—robust, physical pre-frailty, and physical frailty—as outcomes. The logistic regression analysis models included Model 1 without adjusted factors, and Model 2, adjusted for population demographic factors that have been reported to be associated with physical pre-frailty and frailty. The adjustment factors included age; sex; BMI; presence of disease (osteoporosis, hypertension, dyslipidemia, diabetes, stroke, cardiovascular disease, and other diseases); number of pain sites (shoulder, elbow, wrist, hip, knee, ankle, lower back, and neck); Mini-Mental State Examination (MMSE) score; years of education; presence of a tobacco smoking habit (categorized as ‘almost every day’, ‘occasionally’, ‘used to smoke but quit’, or ‘never smoked’; we considered the categories ‘almost every day’ and ‘occasionally’ as having a tobacco smoking habit); alcohol consumption (categorized as ‘almost every day’, ‘occasionally’, ‘rarely’, or ‘never drink’; we considered the categories ‘almost every day’ and ‘occasionally’ as having an alcohol consumption habit); and sedentary time, which was assessed by the Omron Healthcare Active Style Pro HJA-350IT triaxial accelerometer; we considered activity intensity values below 1.5 metabolic equivalent of task (MET) units as sedentary time (ST). These factors were extracted from the questionnaires and measurement results.

## 3. Results

### 3.1. Comparison of Characteristics in the Combination of MVPA and Sleep Quality

The participant response rate among the initially guided population was 930 participants (18.6%), which was low. Furthermore, out of this group, the final analyzable participants comprised 811 participants (87.2%). As shown in Table 1, the number of participants in each group was as follows: 360 individuals (42.2%) in the MVA^+^SLP^+^ group, 105 individuals (12.3%) in the MVA^+^SLP^−^ group, 301 individuals (35.3%) in the MVA^−^SLP^+^ group, and 8Both the MVA^+^SLP^−^ and MVA^−^SLP^−^ groups exhibited a significantly higher number of pain locations. The MVA^−^SLP^+^ group had a higher prevalence of smokers.

The rate of individuals classified as having physical pre-frailty/frailty in each group was as follows: 42.8%/0.6% in the MVA^+^SLP^+^ group, 47.5%/2.0% in the MVA^+^SLP^−^ group, 58.5%/8.2% in the MVA^−^SLP^+^ group, and 65.5%/12.6% in the MVA^−^SLP^−^ group.

### 3.2. Association between the Combination of MVPA and Sleep Quality and Physical Pre-Frailty and Frailty

Using an ordinal logistic regression analysis, we calculated and compared the odds ratios (ORs) and 95% confidence intervals (CIs) for physical pre-frailty and frailty, with the MVA^+^SLP^+^ group serving as the reference group. The MVA^−^SLP^+^ group and the MVA^−^SLP^−^ group showed significantly higher ORs for physical pre-frailty and frailty. As shown in Table 2, this association remained significant even after multivariate adjustment, with an OR of 2.56 (95%CI: 1.80–3.62) in the MVA^−^SLP^+^ group and an OR of 3.97 (95%CI: 2.33–6.74) in the MVA^−^SLP^−^ group.

### 3.3. Association between the Combination of MVPA and Sleep Quality and Components of Physical Frailty

Compared to the MVA^+^SLP^+^ group, the MVA^+^SLP^−^ group showed a significantly higher OR for psychological stress, while the MVA^−^SLP^+^ group had a significantly higher OR for slow gait speed and low physical activity. These associations remained significant after adjustment (Table 3). The MVA^−^SLP^−^ group also had higher ORs for slow gait speed, psychological stress, and low physical activity; the significance of the association for slow gait speed disappeared after adjustment.

## 4. Discussion

The results of our analyses revealed that compared to the group of participants who met the MVPA criteria and had good sleep quality, the prevalence of physical pre-frailty and frailty among the group of participants who did not meet the criteria for MVPA but had good sleep quality was significantly higher, as well as among the group of participants who did not meet the MVPA criteria and had poor sleep quality. In addition, the prevalence of physical frailty in our participants was lower compared to a systematic review that included cross-sectional studies targeting community-dwelling older adults aged ≥65 years (21 papers, total of 61,500 individuals) [28]. Moreover, in comparison to previous studies, we observed a higher proportion of older adults who met the criteria for MVPA, and a similar proportion of older adults with poor sleep quality compared to the reported values [29,30]. In the present participants, objective triaxial accelerometers were used to assess MVPA, which allowed for an accurate calculation of walking and other activities. This difference in assessment methods may have contributed to the discrepancy between the past and present results. When considering sleep quality, it is essential to acknowledge the potential disparities between objectively measured and reported results. Both the previous studies and our study utilized the same PSQI for assessing sleep quality, resulting in similar prevalence rates. However, there is a need to consider the possibility of different proportions when objectively evaluating sleep quality.

As in earlier studies, our present findings demonstrated that individuals who did not meet the criteria for MVPA had a higher prevalence of physical pre-frailty and frailty. Cross-sectional studies have reported a negative association between MVPA and physical frailty in community-dwelling older adults, which aligns with the results of our study [6,15,31]. On the other hand, regarding sleep quality, an investigation of community-dwelling older adults reported a negative association between sleep quality as assessed by the PSQI and physical frailty assessed by the Fried phenotype [10]; however, physical activity was not measured in that study. The novel findings derived from this analysis indicate that when sleep quality is poor, the prevalence of physical pre-frailty and frailty is higher among individuals who do not meet the MVPA criteria. However, if MVPA criteria are met, the prevalence of physical pre-frailty and frailty is not necessarily higher, even when sleep quality is poor.

Although our results also revealed that individuals who met the criteria for MVPA were not independently at a higher prevalence of physical pre-frailty and frailty, even if they had poor sleep quality, the MVA^+^SLP^−^ group had a higher rate of psychological stress. A novel finding of the present study is that individuals who meet the criteria for MVPA are not necessarily at a higher prevalence of physical pre-frailty and frailty, even if the quality of their sleep is poor. An investigation of 2264 community-dwelling older adults indicated that poor sleep quality was associated with daytime fatigue [32], which is consistent with our result, indicating a similar relationship. Although physical activity is effective for preventing frailty, a meta-analysis of randomized controlled trials (RCTs) reported adverse events such as muscle pain, fatigue, and shoulder stiffness in older adults [5]. The possibility that meeting the criteria for MVPA also influenced the increased psychological stress thus cannot be ruled out. Poor sleep quality may also contribute to insufficient performance and muscle recovery, leading to a decrease in the benefits of physical activity [33,34]. In a study of community-dwelling older adults (average age, 82 years), a positive association between sleep quality and physical function was observed, even when the subjects’ average physical activity (counts/min/day) was evaluated by a triaxial accelerometer [13]. We detected no association between sleep quality and physical function measures such as grip strength, gait speed, and weight loss in the present participants, which suggests the possibility that sleep quality does not directly impact performance. The average age of the earlier study’s subjects was higher, and the study’s authors did not consider the influence of activity levels and exercise. The lack of a significant association with the physical pre-frailty and frailty in the MVA^+^SLP^−^ group can be attributed to the prominence of fatigue as the only factor. A study of community-dwelling older adults reported that compared to a group of subjects who had the ‘usual’ sleep pattern of 6–8 h, the long sleep group (≥8 h) were at a higher prevalence of sarcopenia [35]. Our present study included individuals with long sleep durations among those with good sleep quality, and their presence may have influenced the results.

We also observed that the present participants who had good sleep quality but did not meet the criteria for MVPA were at a higher prevalence of physical pre-frailty and frailty. Their group showed a higher prevalence of slow gait speed and low physical activity. MVPA as we defined it is recommended by the WHO to reduce the risk of diseases such as stroke, cardiovascular disease, cancer, and diabetes. Considering that the criteria for MVPA suggested by the WHO typically indicate higher levels of activity compared to the PAEE criteria, it is plausible that both groups meeting the MVPA criteria did not exhibit a higher prevalence of physical frailty due to meeting sufficient PAEE requirements. A positive association between physical activity levels and gait speed has been described [36]. When physical activity levels are low, there is a decrease in high-intensity activities such as walking at a brisk pace and movements requiring lower limb strength in exercise and daily activities. This leads to a decrease in lower limb strength and balance, resulting in a higher risk of slow gait speed and low physical activity [37]. Thus, despite good sleep quality, the association between slow gait speed and not meeting MVPA criteria might suggest a potential link with the prevalence of physical frailty.

Among the present participants, poor sleep quality and a failure to meet the MVPA criteria were associated with a higher prevalence of physical pre-frailty and frailty. When sleep quality declines, the basal metabolic rate decreases, leading to reduced energy expenditure. As energy expenditure decreases, there may be insufficient energy available for maintaining muscle mass and physical functions, resulting in muscle atrophy and decreased muscle strength [1]. Moreover, the reduction in energy expenditure can impact immune function and decrease the body’s resistance, potentially leading to increased susceptibility to illnesses and a subsequent decline in physical function. Additionally, the decline in sleep quality might hinder muscle repair and growth during sleep, potentially contributing to a decrease in muscle mass and strength. Consequently, this decrease in physical activity may further contribute to a reduction in energy expenditure. As mentioned earlier, individuals who do not meet the criteria for MVPA are associated with a higher prevalence of slow gait speed and low physical activity, even if they have good sleep quality, and they are independently at a higher prevalence of physical frailty. Poor sleep quality is associated with a higher rate of psychological stress, which comprehensively contributes to the increased prevalence of physical pre-frailty and frailty.

The strength of this study lies in our consideration of the influence of sleep quality on the association between MVPA and physical pre-frailty and frailty. Another strength of this study is that the participants’ MVPA was assessed using accelerometers and defined based on moderate-to-vigorous physical activity (MPA and VPA), allowing for the objective measurement of activity levels rather than relying on self-report questionnaires.

Several study limitations must be addressed, however. Of the 5000 residents aged 65–75 years in the region invited to participate in this study, the response rate was low at 18.6%. Therefore, there are certain limitations to the generalizability of the research findings. The study’s cross-sectional design prevents any conclusion regarding causality. The longitudinal relationship between the combination of MVPA plus sleep quality and the risk of physical frailty should be examined in future research. Our assessment of sleep quality relied on questionnaires, which may be subject to recall bias; objective measures of sleep quality should be employed in future studies. In this study, we classified physical activity and sleep quality based on cutoffs that have been considered important for health in previous research. However, whether these cutoffs are optimal as indicators of physical frailty remains a subject for future investigation. Both the PAEE, a component of the outcome physical frailty, and the exposure factor MVPA were assessed for activity intensity using the same physical activity monitor. Therefore, one possible explanation for the observation that both groups meeting MVPA criteria did not have a higher prevalence of physical frailty could be attributed to these groups meeting sufficient PAEE requirements. However, meeting the MVPA criteria recommended by the WHO is associated with the criteria for low physical activity (a component of physical frailty), suggesting the potential of meeting the MVPA criteria as a beneficial indicator for older adults. The characteristics of movements in older individuals have shown an underestimation of high-intensity physical activity compared to younger individuals. This suggests the potential presence of participants within the group meeting high MVPA criteria but not categorized as meeting MVPA. However, the authors believe that the results of this study remain consistent within the group meeting MVPA criteria.

## 5. Conclusions

The results of this study demonstrated that, in older adults, not meeting the criteria for MVPA levels is associated with higher odds of physical pre-frailty and frailty, regardless of whether or not their sleep quality was impaired. In the future, it is crucial to comprehend both aspects of physical activity levels and sleep quality among older adults in community settings. Furthermore, there is a need for ongoing investigations to longitudinally examine whether the status of MVPA and sleep quality heightens the risk of physical frailty.

## Figures and Tables

**Figure 1 geriatrics-09-00031-f001:**
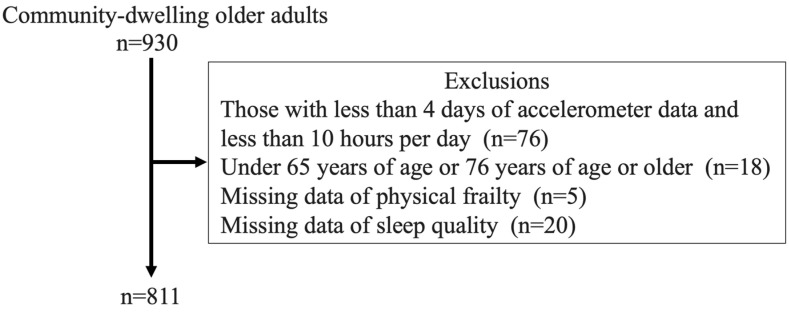
Flowchart of the participants.

**Table 1 geriatrics-09-00031-t001:** Characteristics of MVPA in combination with sleep quality.

	MVA^+^SLP^+^ *n* = 360	MVA^+^SLP^−^ *n* = 105	MVA^−^SLP^+^ *n* = 301	MVA^−^SLP^−^ *n* = 88	*p*-Value
Age, yrs	70 (68–73)	71 (68–73)	70 (68–74)	72 (69–75) *	<0.01
Men	158 (46.3)	42 (41.6)	147 (52.1)	39 (44.8)	0.23
BMI, kg/m^2^	22.6 ± 3.1	22.5 ± 3.2	23.2 ± 3.0	23.2 ± 3.6	0.08
Presence of disease	247 (72.4)	76 (75.3)	227 (80.5)	81 (93.1) *	<0.01
No. of pain sites	1 (0–2)	2 (1–3) *	1 (0–2)	2 (1–3) *	<0.01
Education, yrs	12.9 ± 2.3	12.8 ± 2.5	13.2 ± 2.5	12.5 ± 2.1	0.06
Presence of tobacco smoking habit	18 (5.3)	2 (2.0)	34 (12.1) *	8 (9.2)	<0.01
Presence of alcohol habit	181 (53.1)	52 (51.5)	140 (49.7)	37 (42.5)	0.36
MMSE, score	29 (26–30]	28 (26–30)	28 (26–30)	28 (26–30)	0.55
Sedentary time, min/day	412 ± 104.8	417.8 ± 102.7	486.9 ± 9.9 *	477.1 ± 99.6 *	<0.01
Physical pre-frailty	146 (42.8)	48 (47.5)	165 (58.5) *	57 (65.5) *	<0.01
Physical frailty	2 (0.6)	2 (2.0)	23 (8.2) *	11 (12.6) *	
Low grip strength	63 (18.5)	20 (19.8)	62 (22.0)	23 (26.4)	0.38
Slow gait speed	47 (13.8)	1 (10.9)	74 (26.2) *	27 (31.0) *	<0.01
Psychological stress	28 (8.2)	22 (21.8) *	28 (9.9)	24 (27.6) *	<0.01
Weight loss	33 (9.7)	8 (7.9)	25 (8.9)	11 (12.6)	0.70
Low physical activity	9 (2.6)	2 (2.0)	105 (37.2) *	35 (40.2) *	<0.01

Data are mean ± standard deviation, median (25–75% percentiles), or n (%). * *p* < 0.05 vs. MVPA^+^Sleep^+^. The χ^2^-test was used for comparisons of proportions, the Kruskal–Wallis test for comparisons of medians, and a one-way analysis of variance (ANOVA) for comparisons of means, with respective *p*-values shown. BMI, body mass index; MMSE, Mini-Mental State Examination; MVPA, moderate-to-vigorous physical activity; PSQI, Pittsburgh Sleep Quality Index. MVA^+^SLP^+^, the participants who met the criteria for both MVPA and good sleep quality; MVA^+^SLP^−^, those who met the criteria for MVPA but had poor sleep quality; MVA^−^SLP^+^, those who did not meet the criteria for MVPA but had good sleep quality; MVA^−^SLP^−^, those who did not meet the criteria for both MVPA and sleep quality.

**Table 2 geriatrics-09-00031-t002:** Association between the MVPA and sleep quality combination and physical pre-frailty and frailty.

	*n* (%)	Prevalence, *n* (%)	Model 1			Model 2		
			OR	95%CI	*p*-Value	OR	95%CI	*p*-Value
MVA^+^SLP^+^	360 (42.2)	Pre-frailty, 146 (42.8) Frailty, 2 (0.6)	1.00	reference	−	1.00	reference	−
MVA^+^SLP^−^	105 (12.3)	Pre-frailty, 48 (47.5) Frailty, 2 (2.0)	1.29	0.83–2.01	0.25	1.10	0.70–1.74	0.68
MVA^−^SLP^+^	301 (35.3)	Pre-frailty, 165 (58.5) Frailty, 23 (8.2)	2.86	2.07–3.94	<0.01	2.56	1.80–3.62	<0.01
MVA^−^SLP^−^	88 (10.3)	Pre-frailty, 57 (65.5) Frailty, 11 (12.6)	5.24	3.17–8.69	<0.01	3.97	2.33–6.74	<0.01

Model 1, no adjustment factor; Model 2, adjusted for age, gender, BMI, number of diseases, number of pain sites, MMSE score, smoking habit, alcohol habit, years of education, sedentary time. MVA^+^ SLP^+^, the participants who met the criteria for both MVPA and good sleep quality; MVA^+^SLP^−^, those who met the criteria for MVPA but had poor sleep quality; MVA^−^SLP^+^, those who did not meet the criteria for MVPA but had good sleep quality; MVA^−^SLP^+^, those who did not meet the criteria for both MVPA and sleep quality.

**Table 3 geriatrics-09-00031-t003:** Associations between the MVPA and sleep quality combination and components of physical frailty.

Group	*n* (%)	No. of Participants Applicable to Each Component (The Rate of That Number in Each Group)	Model 1			Model 2		
			OR	95%CI	*p*-Value	OR	95%CI	*p*-Value
Low grip strength								
MVA^+^SLP^+^	360 (42.2)	63 (18.5)	1.00	reference	−	1.00	reference	−
MVA^+^SLP^−^	105 (12.3)	20 (19.8)	1.09	0.62–1.91	0.76	1.02	0.58–1.82	0.94
MVA^−^SLP^+^	301 (35.3)	62 (22.0)	1.24	0.84–1.84	0.28	1.29	0.84–1.99	0.25
MVA^−^SLP^−^	88 (10.3)	23 (26.4)	1.59	0.92–2.75	0.10	1.40	0.77–2.54	0.27
Slow gait speed								
MVA^+^SLP^+^	360 (42.2)	47 (13.8)	1.00	reference	−	1.00	reference	−
MVA^+^SLP^−^	105 (12.3)	1 (10.9)	0.77	0.38–1.54	0.45	0.66	0.32–1.34	0.25
MVA^−^SLP^+^	301 (35.3)	74 (26.2)	2.23	11.48–3.34	<0.01	1.91	1.22–2.99	<0.01
MVA^−^SLP^−^	88 (10.3)	27 (31.0)	2.82	1.63–4.87	<0.01	1.98	1.08–3.61	0.03
Psychological stress								
MVA^+^SLP^+^	360 (42.2)	28 (8.2)	1.00	reference	−	1.00	reference	−
MVA^+^SLP^−^	105 (12.3)	22 (21.8)	3.11	1.69–5.73	<0.01	2.66	1.42–4.96	<0.01
MVA^−^SLP^+^	301 (35.3)	28 (9.9)	1.23	0.71–2.13	0.46	1.26	0.70–2.27	0.43
MVA^−^SLP^−^	88 (10.3)	24 (27.6)	4.26	2.32–7.83	<0.01	3.85	1.98–7.47	<0.01
Weight loss								
MVA^+^SLP^+^	360 (42.2)	33 (9.7)	1.00	reference	−	1.00	reference	−
MVA^+^SLP^−^	105 (12.3)	8 (7.9)	0.80	0.36–1.80	0.59	0.75	0.72–3.45	0.48
MVA^−^SLP^+^	301 (35.3)	25 (8.9)	0.91	0.53–1.57	0.73	1.05	0.33–1.70	0.87
MVA^−^SLP^−^	88 (10.3)	11 (12.6)	1.35	0.65–2.80	0.42	1.57	0.58–1.89	0.26
Low physical activity								
MVA^+^SLP^+^	360 (42.2)	9 (2.6)	1.00	reference	−	1.00	reference	−
MVA^+^SLP^−^	105 (12.3)	2 (2.0)	0.75	0.16–3.51	0.71	0.60	0.12–2.93	0.53
MVA^−^SLP^+^	301 (35.3)	105 (37.2)	21.9	10.82–44.28	<0.01	16.1	7.55–34.23	<0.01
MVA^−^SLP^−^	88 (10.3)	35 (40.2)	24.8	11.28–54.64	<0.01	18.0	7.54–43.08	<0.01

Model 1, no adjustment factor; Model 2, adjusted for age, gender, BMI, number of diseases, number of pain sites, MMSE score, smoking habit, alcohol habit, years of education, sedentary time. MVA^+^ SLP^+^, the participants who met the criteria for both MVPA and good sleep quality; MVA^+^SLP^−^, those who met the criteria for MVPA but had poor sleep quality. MVA^−^SLP^+^, those who did not meet the criteria for MVPA but had good sleep quality; MVA^−^SLP^−^, those who did not meet the criteria for both MVPA and sleep quality.

## Data Availability

Data are contained within the article.

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
