# Peer review of "Association of the Combination of Moderate-to-Vigorous Physical Activity and Sleep Quality with Physical Frailty"

_geriatrics, 2024, doi:10.3390/geriatrics9020031_

Round 1

Reviewer 1 Report (Previous Reviewer 2)

Comments and Suggestions for Authors

I still find the manuscript misleading when the authors talk about risk. Risk cannot be inferred from this study. The direction of association between factors examined in a cross sectional study cannot be ascertained. It could be that sleep and physical activity impact on incidence of frailty, or it could be frailty impacts on sleep and physical activity (and in reality bidirectional relationships are very likely). I previously highlighted this point in relation to the conclusion, but the same principle applies throughout the manuscript. The authors need to improve their manuscript in this regard throughout (especially abstract and discussion) to avoid misleading the reader on this point. 

Author Response

Submission ID geriatrics-2845622: "Association Between the Combination of Moderate-to-Vigorous Physical Activity and Sleep Quality with Physical Frailty"

Response to Reviewer Comment

To the Reviewer: 

We appreciate the opportunity to address your helpful comments and revise our manuscript. Our responses to your comments are given below. The revised parts in the manuscript appear as tracked changes or in red font.

Reviewer 1

Point 1.

I still find the manuscript misleading when the authors talk about risk. Risk cannot be inferred from this study. The direction of association between factors examined in a cross sectional study cannot be ascertained. It could be that sleep and physical activity impact on incidence of frailty, or it could be frailty impacts on sleep and physical activity (and in reality bidirectional relationships are very likely). I previously highlighted this point in relation to the conclusion, but the same principle applies throughout the manuscript. The authors need to improve their manuscript in this regard throughout (especially abstract and discussion) to avoid misleading the reader on this point.

Response 1.

Thank you for your feedback. We apologize for the delay in addressing the issue. As you pointed out, given that this study is a cross-sectional design, the use of the term "risk" might imply a temporal progression in the development of outcomes. Therefore, we have replaced the term "risk" with an alternative in the abstract and discussion sections, and accordingly made necessary adjustments to the text.

Furthermore, we have conducted a general review of our manuscript and made some modifications to the language expression on our own. (Lines 15, 20-24, 207, 338-339)

Reviewer 2 Report (Previous Reviewer 1)

Comments and Suggestions for Authors

Thank you for the response and changes made to the manuscript. It reads much better and all previous concerns have been addressed in an adequate manner.

Author Response

I would like to express my sincere gratitude for your valuable advice throughout the review process. Thanks to your insights, I am confident that this paper has improved significantly. I look forward to incorporating your feedback into future research endeavors. Thank you very much for your invaluable contributions.

This manuscript is a resubmission of an earlier submission. The following is a list of the peer review reports and author responses from that submission.

Round 1

Reviewer 1 Report

Comments and Suggestions for Authors

I find this article highly interesting and relevant. Physical activity and sleep highly interact with each other, and looking closer into the relationship between these two are important to gain more knowledge about how these interact in older adults.

All in all, I find the article to be well written. That said there are some details that I think the authors need to clarify before publication.

-          When measuring activity with accelerometers it is highly dependent on how the data is analyzed. As far as I understand the authors have used the same method as Ohkawara et al (2011)? As far as I know this is an accelerometer that comes with a included algorithm to get out activity data? Can you please elaborate somewhat on the methodology of the accelerometer and algorithm, and possible post processing of the data? Also, as far as I know this is not validated on older adults? This should be discussed as we know that there are differences on how adults and older adults move and the amount of energy they use on different activities.

-          Even though PA is measured objectively in this study, sleep is measured using a questionnaire. The authors mention this as a limitation in the end, but I miss that they take this into account in the discussion for instance on p.8, first paragraph. The differences between these findings and previous findings would indeed also be affected by how sleep quality is measured. Objectively measured sleep is more and more prevalent, and I miss the inclusion of this aspect as well in the article.

-          In table 3 there is a column named “No. of applicable persons”, which include a significant lower number of participants than the included n. What is this? I cant seem to find any explanation of this column in the methods. Please elaborate and describe. Why are these applicable? And why do you loose so many? Also, if this is the case, this needs to be discussed.

-          The name of the measurement groups are somewhat unfortunate. “Shrinking” and “slowness” is not a good description of people, nor a very good academic description. I would recommend changing this into for instance weight loss and “reduced gait speed” which would be more relatable with previous research. Also “exhaustion” is often related to physical exhaustion as measured with for instance Borg-scale, but as far as I understand the current measure is more related to psychological stress. Hence, I would also recommend changing this name.

-          You have a large cohort to include from. As you state, you have a higher portion of older adults that sleep well and meet the PA criteria. Do you think there is a bias in who signed up for this project of the 5000 invited? Do you know anything about the over 4000 people that did not join so that you possible could describe your selection at baseline, such as age, gender, physical function? I think that would be an interesting aspect to add in methods and describe as it might provide additional information about your population.

Author Response

Submission ID geriatrics-2737683: "Association Between the Combination of Moderate-to-Vigorous Physical Activity and Sleep Quality with Physical Frailty"

Response to Reviewer Comments

To the Reviewer: 

We appreciate the opportunity to address your helpful comments and revise our manuscript. Below are our item-by-item responses to your comments. The revised parts in the manuscript appear as tracked changes and in red font.

Reviewer 1

Point 1:

When measuring activity with accelerometers it is highly dependent on how the data is analyzed. As far as I understand the authors have used the same method as Ohkawara et al (2011)? As far as I know this is an accelerometer that comes with a included algorithm to get out activity data? Can you please elaborate somewhat on the methodology of the accelerometer and algorithm, and possible post processing of the data? Also, as far as I know this is not validated on older adults? This should be discussed as we know that there are differences on how adults and older adults move and the amount of energy they use on different activities.

Response 1:

Line 137,

The study by Ohkawara et al. (2011) targeted individuals in middle age, which differs from the participants in our study. Therefore, we shifted to a prior study that demonstrated the validity of an accelerometer among older adults.

“25) Nagayoshi S, Oshima Y, Ando T, Aoyama T, Nakae S, Usui C, Kumagai S, Tanaka S. Validity of estimating physical activity intensity using a triaxial accelerometer in healthy adults and older adults. BMJ Open Sport Exerc Med. 2019;5:e000592. doi: 10.1136/bmjsem-2019-000592.”

I have added the following text regarding the activity monitor's algorithm and post-processing of data.

Line 117-127,

“The accelerometer is a device that continuously measures exercise intensity using an algorithm derived from the relationship between data obtained by an acceleration sensor and energy consumption determined by expired gas analysis. Post-processing of the data followed established procedures from previous studies [21]. Anteroposterior (x-axis), mediolateral (y-axis), and vertical (z-axis) acceleration measurements were collected during each activity at a rate of 32 Hz with 12-bit accuracy. The acceleration data range for each axis is ±6 G, resulting in a resolution of 3 mG. Each signal from the triaxial accelerometer underwent high-pass filtering with a cutoff frequency of 0.7 Hz to eliminate the gravitational acceleration component. The integral of the absolute value of the acceleration signals from all three axes was calculated over 10-second intervals.”

Due to differences in movement between older and younger individuals, there is variability in energy expenditure during physical activities. As a result, I have added the following text to the limitations section:

Line 359-363,

"The characteristics of movements in older individuals have shown an underestimation of high-intensity physical activity compared to younger individuals. This suggests the potential presence of participants within the group meeting high MVPA criteria but not categorized as meeting MVPA. However, the authors believe that the results of this study remain consistent within the group meeting MVPA criteria."

Point 2:

Even though PA is measured objectively in this study, sleep is measured using a questionnaire. The authors mention this as a limitation in the end, but I miss that they take this into account in the discussion for instance on p.8, first paragraph. The differences between these findings and previous findings would indeed also be affected by how sleep quality is measured. Objectively measured sleep is more and more prevalent, and I miss the inclusion of this aspect as well in the article.

Response 2:

We only demonstrated the differences between objective and subjective measurements using the accelerometer. As you mentioned, there are studies concerning the objective measurement of sleep quality. Hence, I have added the following considerations regarding sleep quality to the discussion section:

Line 255-259,

"When considering sleep quality, it is essential to acknowledge the potential disparities between objectively measured and reported results. Both previous studies and our study utilized the same PSQI for assessing sleep quality, resulting in similar prevalence rates. However, there is a need to consider the possibility of different proportions when objectively evaluating sleep quality."

Point 3:

In table 3 there is a column named “No. of applicable persons”, which include a significant lower number of participants than the included n. What is this? I cant seem to find any explanation of this column in the methods. Please elaborate and describe. Why are these applicable? And why do you loose so many? Also, if this is the case, this needs to be discussed.

Response 3:

Table 3,

The 'No' in Table 3 represents the number of participants (percentage within that group) with each component of physical frailty. We added this explanation as it was previously lacking.

Point 4:

The name of the measurement groups are somewhat unfortunate. “Shrinking” and “slowness” is not a good description of people, nor a very good academic description. I would recommend changing this into for instance weight loss and “reduced gait speed” which would be more relatable with previous research. Also “exhaustion” is often related to physical exhaustion as measured with for instance Borg-scale, but as far as I understand the current measure is more related to psychological stress. Hence, I would also recommend changing this name.

Response 4:

As you suggested, some terms might lead to misunderstandings. Therefore, we have changed 'Weakness' to 'Low grip strength,' 'Slowness' to 'Slow gait speed,' 'Exhaustion' to 'Psychological stress,' and 'Shrink' to 'Weight loss' in the entire text and Table 3."

Point 5:

You have a large cohort to include from. As you state, you have a higher portion of older adults that sleep well and meet the PA criteria. Do you think there is a bias in who signed up for this project of the 5000 invited? Do you know anything about the over 4000 people that did not join so that you possible could describe your selection at baseline, such as age, gender, physical function? I think that would be an interesting aspect to add in methods and describe as it might provide additional information about your population.

Response 5:

Line 73-74,

Indeed, comparing the backgrounds of participants and non-participants in this study provides important information. However, the number of participants in this investigation was limited to only 930 individuals. Over 4000 people received the invitation by mail but did not participate in this study, so we lack information about them.

Reviewer 2 Report

Comments and Suggestions for Authors

Thank you for asking me to review Association between the combination of MVPA and sleep quality with physical frailty

Overall this is a very well written and considered study. Although limited by the cross sectional design it is interesting and worthy of publication. It should help inform future work and hopefully will lead to prospective evaluations of the relationships presented. 

Abstract- well written. Contains all the pertinent points. 

Introduction – well written and concise. Supports the hypothesis to be tested. 

Methods- overall very clear. Minor point:

1)    Were participants allowed to use walking aids during walking speed assessment?

2)    Were analyses adjusted for accelerometer wear time? If not, could the authors justify?

Results- I am usually very critical of the results but I find the results presented transparently and with easy to follow logic. 

Discussion

The interaction between sleep quality and MVPA is an interesting finding. It may not necessarily mean that individuals with poor sleep quality who meet MVPA criteria are NOT at increased odds of frailty compared to those that do not have poor sleep quality. It may be that the higher odds reported was too small an effect size for this study to detect. This should be discussed. I certainly agree that there is evidence the two risk factors may potentiate each other, and when both present magnify the effect size seen. 

The discussion would benefit from consideration that results presented are cross sectional. Thus, when direction of association is implied it must be made very clear that this cannot be deduced from this work. For example, line 291-293 ‘The risk of slow gait speed is thus strongly influenced by an individuals failure to meet the criteria for MVPA, even if his or her sleep quality is good’. This suggests the authors are referring to the results presented rather than assimilating inferences from wider literature including prospective studies. This study cannot evaluate direction of association and thus cannot fully support this statement as strongly as it is worded. The authors should carefully revise their discussion to avoid inferring the results presented identify causal relationships. 

One of the most interesting findings is the interaction between sleep quality and MVPA. It would be helpful if authors could discuss possible underlying biological mechanisms explaining this.i.e., is this a biologically plausible interaction? They have started in lines 294-301. However, it would be interesting to expand this. For example, given the finding of consistent associations between poor sleep quality and exhaustion, it may be that sleep impacts on the evolution of frailty at an earlier time point than physical activity. (considering the fried frailty cycle of reduced energy expenditure leading to reduced strength and mobility, then weight loss and further lowering of resting metabolic rate). 

Please remove claims of novelty—line 305 ‘…, but our present study is the first…..’ It is not possible to be certain this is the case. 

I do not understand lines 316-320 as written. I think they are trying to explain the problem of physical activity, here an exposure variable, also being a component of the outcome, physical frailty. This is a very real problem, faced by others investigating physical activity/ frailty relationships. I think the authors have handled this well in their analyses and have been transparent in examining relationships between exposures and the components of the fried frailty phenotype, in addition to pre-frailty and frailty as a whole construct. However, this limitation needs very clear consideration in the discussion, and it is currently not well explained. 

Conclusions

The authors cannot talk about ‘risk’. The study is cross sectional and direction of causality cannot be inferred. Use of the term ‘risk’ implies a cause and effect. A better statement is that not meeting the criteria for MVPA levels is associated with higher odds of frailty, regardless of sleep quality. 

Within the conclusions it would be helpful to indicate what future work is required to further explore the relationships presented.

Reviewer 3 Report

Comments and Suggestions for Authors

Interesting idea for the study. Please consider suggestions and questions.

Author Response

Submission ID geriatrics-2737683: "Association Between the Combination of Moderate-to-Vigorous Physical Activity and Sleep Quality with Physical Frailty"

Response to Reviewer Comments

To the Reviewer: 

We appreciate the opportunity to address your helpful comments and revise our manuscript. Our item-by-item responses to your comments are given below. The revised parts in the manuscript appear as tracked changes or in red font.

Reviewer 3

Point 1:

Line 14 -16- confusing statement. I think you are trying to say they were not receiving nursing care, etc. May need to reword.

Response 1:

Line 14-16,

I have revised the text as follows.

“A cross-sectional study was conducted with a participants pool comprising older adults who had not required nursing care or support services.”

Point 2.

Line 17- 19 needs to be better worded and acronyms spelled out.

Response 2:

We have added the formal names for 'MVPA' and 'VPA'.

Point 3:

Page 3 line 90 shrinking implies getting shorter rather that losing weight. A reference to support this would be good. I did not see where Liu and Fried supported this statement.

Response 3:

The term 'Shrinking' was unclear, so we changed it to 'Weight loss'.

Point 4:

line 101- what do you mean the arm hanging down. Is this the proper technique for assessment?

Response 4:

Line 101-102,

The term 'hanging arms' refers to a posture in a natural standing position where there is no bending of shoulders or elbows while standing. This posture represents a common and appropriate method for assessing grip strength using a Smedley-type hand dynamometer.

Point 5:

014- why not use the formal gait speed test? Also was their only one trial or multiple?

Response 5:

Line 109-110,

While evaluating gait speed, we conducted the assessment at usual walking speed, a common practice in frailty assessment. Additionally, gait speed was assessed twice, and the faster speed was recorded. I have added this explanation as it was previously lacking.

Point 6: Did you train the participant in the use or wearing of the accelerometer?

Response 6:

Line 127-129

The researchers explained the method of wearing the activity monitor to each participant individually, both in written instructions and verbally, and assisted the participants in putting on the device.

Point 7:

Overall – I worry about the methods and the development of the criteria for the groupings. Otherwise the results and conclusion are fine. More robust methods.

Response 7:

Line 349-351,

In this study, we classified physical activity and sleep quality based on cutoffs that have been considered important for health in previous research. However, whether these cutoffs are optimal as indicators for physical frailty remains a subject for future investigation, a point that has been added to the limitations of this study.